# Secular trends in incidence of type 1 and type 2 diabetes in Hong Kong: A retrospective cohort study

Andrea O. Y. Luk[1,2,3]*, Calvin Ke[1,4], Eric S. H. Lau[1], Hongjiang Wu[1], William Goggins[5], Ronald C. W. Ma[1,2,3], Elaine Chow[1], Alice P. S. Kong[1,2,3], Wing-Yee So[1,6], Juliana C. N. Chan[1,2,3]

1 Department of Medicine and Therapeutics, The Chinese University of Hong Kong, Prince of Wales Hospital, Hong Kong Special Administrative Region, People's Republic of China, 2 Hong Kong Institute of Diabetes and Obesity, Prince of Wales Hospital, The Chinese University of Hong Kong, Hong Kong Special Administrative Region, People's Republic of China, 3 Li Ka Shing Institute of Health Sciences, The Chinese University of Hong Kong, Hong Kong Special Administrative Region, People's Republic of China, 4 Department of Medicine, University of Toronto, Toronto, Canada, 5 School of Public Health, The Chinese University of Hong Kong, Prince of Wales Hospital, Hong Kong Special Administrative Region, People's Republic of China, 6 Hong Kong Hospital Authority, Hong Kong Special Administrative Region, People's Republic of China

* andrealuk@cuhk.edu.hk

**Data Availability Statement:** The data for this study is hosted by the Hong Kong Hospital Authority. Subject to local law and regulation regarding the use and distribution personal data,

## Abstract

### Background

There is very limited data on the time trend of diabetes incidence in Asia. Using population-level data, we report the secular trend of the incidence of type 1 and type 2 diabetes in Hong Kong between 2002 and 2015.

### Methods and findings

The Hong Kong Diabetes Surveillance Database hosts clinical information on people with diabetes receiving care under the Hong Kong Hospital Authority, a statutory body that governs all public hospitals and clinics. Sex-specific incidence rates were standardised to the age structure of the World Health Organization population. Joinpoint regression analysis was used to describe incidence trends.

A total of 562,022 cases of incident diabetes (type 1 diabetes [$n = 2,426$]: mean age at diagnosis is 32.5 years, 48.4% men; type 2 diabetes [$n = 559,596$]: mean age at diagnosis is 61.8 years, 51.9% men) were included. Among people aged <20 years, incidence of both type 1 and type 2 diabetes increased. For type 1 diabetes, the incidence increased from 3.5 (95% CI 2.2–4.9) to 5.3 (95% CI 3.4–7.1) per 100,000 person-years (average annual percentage change [AAPC] 3.6% [95% CI 0.2–7.1], $p < 0.05$) in boys and from 4.3 (95% CI 2.7–5.8) to 6.4 (95% CI 4.3–8.4) per 100,000 person-years (AAPC 4.7% [95% CI 1.7–7.7], $p < 0.05$] in girls; for type 2 diabetes, the incidence increased from 4.6 (95% CI 3.2–6.0) to 7.5 (95% CI 5.5–9.6) per 100,000 person-years (AAPC 5.9% [95% CI 3.4–8.5], $p < 0.05$) in boys and from 5.9 (95% CI 4.3–7.6) to 8.5 (95% CI 6.2–10.8) per 100,000 person-years (AAPC 4.8% [95% CI 2.7–7.0], $p < 0.05$) in girls. In people aged 20 to <40 years, incidence

the database used in the present study cannot be deposited in public data repositories. Data can be applied through the Data Sharing Portal of the Hong Kong Hospital Authority (https://www3.ha.org.hk/data/DCL/Index/).

**Funding:** The authors received no specific funding for this work.

**Competing interests:** RCWM acknowledges receiving research support (outside of this work) from AstraZeneca, Bayer, Pfizer for conducting clinical trials and honoraria or consultancy fees from AstraZeneca and Boehringer Ingelheim, all of which has been donated to the Chinese University of Hong Kong to support diabetes research. RCWM is a member of the Editorial Board of PLoS Medicine. AOYL acknowledges receiving research support (outside of this work) from Boehringer Ingelheim, MSD, Sanofi, Amgen and travel grants from travel grant from AstraZeneca, Boehringer Ingelheim, MSD, Novartis, Novo Nordisk, Sanofi. JCNC and RCWM are cofounders of GemVCare, a diabetes genetic testing laboratory, which was established through support from the Technology Start-up Support Scheme for Universities (TSSSU) from the Hong Kong Government Innovation and Technology Commission (ITC).

**Abbreviations:** AAPC, average APC; APC, annual percentage change; EURODIAB, European Diabetes; FPG, fasting plasma glucose; HA, Hospital Authority; HbA1c, glycated haemoglobin; HKDR, Hong Kong Diabetes Register; HKDSD, Hong Kong Diabetes Surveillance Database; ICD-9, *International Classification of Diseases, Ninth Revision*; NPV, negative predictive value; OGTT, oral glucose tolerance test; PPV, positive predictive value.

of type 1 diabetes remained stable, but incidence of type 2 diabetes increased over time from 75.4 (95% CI 70.1–80.7) to 110.8 (95% CI 104.1–117.5) per 100,000 person-years (AAPC 4.2% [95% CI 3.1–5.3], $p < 0.05$) in men and from 45.0 (95% CI 41.4–48.6) to 62.1 (95% CI 57.8–66.3) per 100,000 person-years (AAPC 3.3% [95% CI 2.3–4.2], $p < 0.05$) in women. In people aged 40 to <60 years, incidence of type 2 diabetes increased until 2011/2012 and then flattened. In people aged ≥60 years, incidence was stable in men and declined in women after 2011. No trend was identified in the incidence of type 1 diabetes in people aged ≥20 years. The present study is limited by its reliance on electronic medical records for identification of people with diabetes, which may result in incomplete capture of diabetes cases. The differentiation of type 1 and type 2 diabetes was based on an algorithm subject to potential misclassification.

## Conclusions

There was an increase in incidence of type 2 diabetes in people aged <40 years and stabilisation in people aged ≥40 years. Incidence of type 1 diabetes continued to climb in people aged <20 years but remained constant in other age groups.

## Author summary

### Why was this study done?

- Diabetes affects over 400 million people worldwide, and over half of the diabetes population comes from Asian countries.

- Most studies on the burden of diabetes in Asia reported the diabetes prevalence, i.e., the proportion of the population with disease, but few have considered diabetes incidence, i.e., the rate at which new cases have developed in the population.

- Knowledge of disease incidence informs how population exposure to risk factors has changed over time and is useful for projection of future prevalence.

### What did the researchers do and find?

- We identified 562,022 people with new-onset type 1 or type 2 diabetes occurring between 2002 and 2015 in the electronic medical record system of the Hong Kong Hospital Authority.

- We calculated the incidence rates of diabetes according to age categories and gender and analysed incidence trends over time.

- We found that the incidence of both type 1 and type 2 diabetes increased in children and adolescents (aged <20 years).

- Incidence of type 2 diabetes also increased in people aged 20 to <40 years but was stable in people aged ≥40 years.

**What do these findings mean?**

- Our results raised concerns over the escalating burden of young-onset diabetes in Asia.

## Introduction

Type 2 diabetes mellitus is a complex progressive disease of rapidly growing prevalence. The global prevalence of type 2 diabetes is currently estimated at 415 million and is projected to escalate to 642 million over the next 25 years [1]. More than half of the world's population of those with type 2 diabetes comes from Asia, where rapid industrialisation and urbanisation have contributed to an obesogenic living environment and increasing rates of overweight and obesity [1].

The majority of published population-based studies on the epidemiology of type 1 and type 2 diabetes in Asia, including China, reported prevalence data, but few examined the incidence of diabetes [2,3]. Although the prevalence informs the disease burden, the incidence reflects how population exposure to risk factors has changed over time, and these estimates are necessary for the accurate projection of future prevalence. Importantly, disease incidence communicates to the health policy makers whether strategies to prevent diabetes have been effective. Recently, Quan and colleagues reported the incidence and prevalence of diabetes in Hong Kong and detected a decline in the incidence of diabetes over a 9-year period between 2006 and 2014 [4]. However, diabetes types were not differentiated, and trends among the paediatric population were not explored. We accessed the territory-wide database hosted by the Hong Kong Hospital Authority (HA) and described the secular trends in incidence of type 1 and type 2 diabetes from 2002 to 2015.

## Methods

### Setting

Hong Kong is a special administrative region of People's Republic of China and has a population of 7.44 million. The Hong Kong HA is a statutory body formed in 1996 that governs all 47 public hospitals and 73 government outpatient clinics in Hong Kong. Because of the wide cost differential between public and private healthcare sectors, around 89% of the local residents receive care for chronic illnesses in the HA [5]. A territory-wide electronic medical record system, adopted in 2000, captures demographic information, diagnostic and procedure codes, laboratory results, and drug prescriptions of people attending public hospitals and clinics.

### Study population

The Hong Kong Diabetes Surveillance Database (HKDSD) is a population-based cohort of people with diabetes in Hong Kong identified from the HA electronic medical record system between 1 January 2000 and 31 December 2016. The analysis was not prespecified and was planned after obtaining and reviewing the content of the database. Diabetes was ascertained based on one or more of the following qualifying criteria: (1) glycated haemoglobin (HbA1c) $\geq$ 6.5% (48 mmol/mol) in any one available HbA1c measurement [6]; (2) fasting plasma glucose (FPG) $\geq$ 7.0 mmol/L in any one available FPG measurement [7]; (3) prescription of noninsulin antihyperglycaemic drugs and/or (4) prescription of insulin for at least 28 days continuously, with or without (5) recording of the diagnostic code of diabetes based on the

*International Classification of Diseases, Ninth Revision* (ICD-9) code 250; and/or (6) recording of the diagnostic code of diabetes according to the revised edition of the *International Classification of Primary Care, World Organization of National Colleges, Academics, and Academic Associations of General Practitioners/Family Physicians* code T89 or code T90. To minimise misclassification of normal individuals as having diabetes, people who received diagnostic coding of diabetes but did not fulfil any of the laboratory or drug criteria throughout the observation are not considered. To avoid inclusion of women with gestational diabetes, we removed episodes occurring within 9 months prior to or 6 months after delivery (ICD-9 codes 72–75) or occurring within 9 months before or after any pregnancy-related encounter (ICD-9 codes 630–676). However, women with subsequent episodes that met the criteria for diabetes occurring outside the context of any obstetric events would be included.

The separation of diabetes types is not reliable based on coding alone in administrative databases. In the HKDSD, a small subset of people received ICD-9 coding for both type 1 and type 2 diabetes. For the purpose of this analysis, we developed and validated an algorithm to delineate type 1 from type 2 diabetes using another database, the Hong Kong Diabetes Register (HKDR). In brief, the HKDR contains clinical information on people with physician-diagnosed diabetes who were referred to two public hospitals (Prince of Wales Hospital and Alice Ho Miu Ling Nethersole Hospital) for assessment of diabetes complications. In the HKDR, diabetes subtype was determined by the referring physician and was independently confirmed with chart review by one of the investigators (CK) of the present study. Type 1 diabetes was defined as clinical presentation with diabetic ketoacidosis and/or continuous requirement of insulin within 1 year of diagnosis. Positivity for islet autoantibodies was not used to define autoimmune diabetes because antibodies were not routinely measured. Of 24,060 patients in the HKDR, we excluded people with onset dates outside of 1 January 2002 to 31 December 2015 (*n* = 8,403) and those with missing data on diabetes subtype (*n* = 357). Of the remaining 15,297 people with newly diagnosed diabetes, type 1 diabetes was confirmed in 103 patients. The cohort was separated in a 2:1 ratio for derivation and validation of the algorithm, respectively. We considered ICD-9 codes (type 1 diabetes: ICD-9 250.x1 or 250.x3, type 2 diabetes: ICD-9 250.x0 or 250.x2) and types of insulin treatment in the algorithms, and we tested 12 combinations for sensitivity, specificity, positive predictive value (PPV) and negative predictive value (NPV) for identifying type 1 diabetes. Of these combinations, ratios of type 1 to type 2 diabetes codes ≥4 or prescription of a combination of short- and long-acting insulin and no noninsulin antihyperglycaemic treatment within the first year of diagnosis yielded the optimal sensitivity of 100.0% (95% CI 76.8–100.0), specificity of 100.0% (95% CI 87.2–100.0), PPV of 100.0% (95% CI 76.8–100.0), and NPV of 100.0% (95% CI 87.2–100.0) in the HKDR for people aged <20 years in the validation cohort. The sensitivity and PPV of this algorithm for identifying type 1 diabetes decreased to 50.0%–71.4% and 28.6%–62.5%, respectively, in people aged ≥20 years, which is expected because of the rarity of type 1 diabetes at older ages.

## Statistical analysis

An incident case of diabetes was identified by the first occurrence of any episode fulfilling the definition of diabetes and at least 2 years of diabetes-free observation prior. At least 1 year of surveillance from the date of diagnosis was required to enable discrimination of type 1 and type 2 diabetes. Thus, although the database includes cases of diabetes from 1 January 2000 to 31 December 2016, incidence of diabetes was described from 1 January 2002 to 31 December 2015. The date of diagnosis was the date to first fulfil the qualifying event. We estimated sex-specific annual incidence rates of type 1 and type 2 diabetes. The numerator was the number of incident cases of diabetes in the HKDSD in each calendar year, and the denominator was

the estimated Hong Kong population of the previous year at midyear as reported by the local census and statistics department. Calculated rates were expressed per 100,000 person-years. We reported age-standardised rates (using the age structure of the World Health Organization standard population) for the entire population and for subgroups by sex and age categories (<20 years, 20 to <40 years, 40 to <60 years, ≥60 years). We calculated the annual percentage change (APC) and the average APC (AAPC) in incidence rates with 95% CI by sex and by age bands. Joinpoint regression analysis was used to describe incidence trends over time. The optimal number of line segments that best fit the pattern was identified, and APCs were computed for the slope before and after each joinpoint using regression analysis. Because of an observed irregularity in the number of new cases of type 2 diabetes in 2004, trend analysis including reporting of APC and AAPC for both type 2 and type 1 diabetes (for consistency) was restricted to the period between 1 January 2005 and 31 December 2015. In a sensitivity analysis, we conducted trend analysis for type 2 diabetes using restricted cubic spline, which allowed for a more flexible fitting of the change in incidence rates, and results are presented in the Supporting information. Statistical significance was set at $p$-value of <0.05. Analysis was performed using SAS version 9.4 (Cary, NC) and Joinpoint Regression Program version 4.6.0.0 (National Cancer Institute, Bethesda, MD).

## Results

Of 778,051 people captured in the HKDSD, we excluded 33,916 people who had diabetes codes but did not fulfil other criteria of diabetes. We further excluded 137,569 cases of prevalent diabetes (episodes fulfilling diabetes between 1 January 2000 and 31 December 2001) and 44,544 cases of diabetes that occurred between 1 January 2016 and 31 December 2016. The remaining 562,022 people with incident type 1 or type 2 diabetes occurring between 2002 and 2015 were included in the analysis (S1 Fig, S1 Table). Demographic and clinical characteristics of the cohort at diagnosis are detailed in S2 Table.

### Trends in incidence of type 1 diabetes

Using the derived algorithm, 2,426 people with newly diagnosed diabetes were classified as having type 1 diabetes, among whom 774 (31.9%) were aged <20 years and 845 (34.8%) and 807 (33.3%) were aged 20 to <40 years and ≥40 years, respectively (S1 Table). Age-standardised incidence of type 1 diabetes increased from 2005 to 2015 in people aged <20 years in both sexes, from 3.5 (95% CI 2.2–4.9) to 5.3 (95% CI 3.4–7.1) per 100,000 person-years (AAPC 3.6% [95% CI 0.2–7.1], $p < 0.05$) in boys and from 4.3 (95% CI 2.7–5.8) to 6.4 (95% CI 4.3–8.4) per 100,000 person-years (AAPC 4.7% [95% CI 1.7–7.7], $p < 0.05$) in girls (Table 1). Broken down into narrower age bands, the incidence of type 1 diabetes peaked in the age groups of 5–9 years in the female sex and 10–14 years in the male sex and declined with increasing age. Among those aged 20 to <40 years and ≥40 years, incidence of type 1 diabetes remained stable over time (Table 1). Sex differences in the risk of type 1 diabetes were detected in the people aged <20 years but not in people aged ≥20 years. The rate ratio for type 1 diabetes was 1.38 (95% CI 1.10–1.37, $p < 0.05$) in girls compared with boys.

### Trends in incidence of type 2 diabetes

Among 559,596 incident cases of type 2 diabetes, 1,182 (0.2%) presented below 20 years of age, 22,924 (4.1%) presented in people aged 20 to <40 years, and 535,490 presented (95.7%) in people aged ≥40 years (S1 Table). The incidence rose with age and peaked in the age groups of 65–75 years in both men and women (Fig 1). A transient increase in the number of incident cases of type 2 diabetes was recorded in 2004. The number of new cases was 34,921 in 2002,

**Table 1. Joinpoint analysis of the trend in incidence of type 1 and type 2 diabetes in Hong Kong Chinese stratified by sex and age groups between 2005 and 2015.**

| Age and sex categories | 2005–2015 | | Time period 1 | | | Time period 1 | | |
|---|---|---|---|---|---|---|---|---|
| | AAPC (95% CI) | p | Year | APC (95% CI) | p | Year | APC (95% CI) | p |
| Type 1 diabetes | | | | | | | | |
| Age <20 years | | | | | | | | |
| Boys | 3.6 (0.2–7.1) | <0.05 | 2005–2013 | 1.5 (−4.7 to 8.0) | 0.59 | 2013–2015 | 17.3 (−30.4 to 97.5) | 0.48 |
| Girls | 4.7 (1.7–7.7) | <0.05 | 2005–2008 | 8.7 (−11.9 to 34.2) | 0.37 | 2008–2015 | 3.6 (−1.8 to 9.2) | 0.16 |
| Age 20 to <40 years | | | | | | | | |
| Men | −1.8 (−6.1 to 2.7) | 0.39 | 2005–2015 | −1.8 (−6.1 to 2.7) | 0.39 | | | |
| Women | −0.8 (−5.7 to 4.3) | 0.72 | 2005–2015 | −0.8 (−5.7 to 4.3) | 0.72 | | | |
| Age 40 to <60 years | | | | | | | | |
| Men | −1.7 (−6.7 to 3.6) | 0.48 | 2005–2015 | −1.7 (−6.7 to 3.6) | 0.48 | | | |
| Women | 1.1 (−5.5 to 8.2) | 0.72 | 2005–2015 | 1.1 (−5.5 to 8.2) | 0.72 | | | |
| Age ≥60 years | | | | | | | | |
| Men | 1.2 (−6.1 to 9.1) | 0.73 | 2005–2015 | 1.2 (−6.1 to 9.1) | 0.73 | | | |
| Women | −2.2 (−9.8 to 6.0) | 0.55 | 2005–2015 | −2.2 (−9.8 to 6.0) | 0.55 | | | |
| Type 2 diabetes | | | | | | | | |
| Age <20 years | | | | | | | | |
| Boys | 5.9 (3.4–8.5) | <0.05 | 2005–2008 | 0.8 (−15.1 to 19.6) | 0.92 | 2008–2015 | 7.5 (2.9–12.4) | <0.05 |
| Girls | 4.8 (2.7–7.0) | <0.05 | 2005–2008 | −2.1 (−15.5 to 13.4) | 0.73 | 2008–2015 | 7.1 (3.0–11.2) | <0.05 |
| Age 20 to <40 years | | | | | | | | |
| Men | 4.2 (3.1–5.3) | <0.05 | 2005–2008 | 8.8 (4.8–13.0) | <0.05 | 2005–2015 | 2.2 (1.3–3.2) | <0.05 |
| Women | 3.3 (2.3–4.2) | <0.05 | 2005–2010 | 4.6 (2.8–6.4) | <0.05 | 2010–2015 | 2.0 (0.4–3.6) | <0.05 |
| Age 40 to <60 years | | | | | | | | |
| Men | 2.1 (0.7–3.6) | <0.05 | 2005–2012 | 3.9 (2.4–5.4) | <0.05 | 2012–2015 | −1.9 (−6.7 to 3.1) | 0.37 |
| Women | 0.5 (−1.4 to 2.3) | 0.63 | 2005–2011 | 2.7 (0.1–5.3) | <0.05 | 2011–2015 | −2.8 (−6.9 to 1.5) | 0.16 |
| Age ≥60 years | | | | | | | | |
| Men | −0.1 (−1.8 to 1.6) | 0.88 | 2005–2015 | −0.1 (−1.8 to 1.6) | 0.88 | | | |
| Women | −1.7 (−3.9 to 0.5) | 0.12 | 2005–2011 | 1.1 (−1.8 to 4.2) | 0.39 | 2011–2015 | −5.8 (−10.6 to −0.8) | <0.05 |

Abbreviations: AAPC, average annual percentage change; APC, annual percentage change

37,229 in 2003, 58,375 in 2004, 33,759 in 2005, and 31,061 in 2006 (S1 Table, S2 Fig). The majority (76.2%) of the new episodes of type 2 diabetes in 2004 was identified by first-time use of noninsulin antihyperglycaemic drugs alone, suggesting that these occurrences could be prevalent cases initially diagnosed and followed outside of HA and later presenting to HA for continuation of existing treatment. Incident cases captured in 2004 or before were excluded from subsequent trend analysis.

An increase in the incidence of type 2 diabetes was observed in the <20 years and 20 to <40 years age groups (Table 1, Fig 2A and 2B). For people <20 years old, the age-standardised incidence of type 2 diabetes increased from 4.6 (95% CI 3.2–6.0) to 7.5 (95% CI 5.5–9.6) per 100,000 person-years (AAPC 5.9% [95% CI 3.4–8.5], $p < 0.05$) in boys and from 5.9 (95% CI 4.3–7.6) to 8.5 (95% CI 6.2–10.8) per 100,000 person-years (AAPC 4.8% [95% CI 2.7–7.0], $p < 0.05$) in girls (Table 1) from 2005 to 2015. For people 20 to <40 years old, we identified a greater increase in the incidence of type 2 diabetes before 2010 in women and before 2008 in men, after which the ascending trend slowed down. In men, the incidence increased from 75.4 (95% CI 70.1–80.7) to 94.4 (95% CI 88.3–100.5) per 100,000 person-years (APC 8.8% [95% CI 4.8–13.0], $p < 0.05$) between 2005 and 2008 and from 94.4 (95% CI 88.3–100.5) to 110.8 (95% CI 104.1–117.5) per 100,000 person-years (APC 2.2% [95% CI 1.3–3.2], $p < 0.05$) between

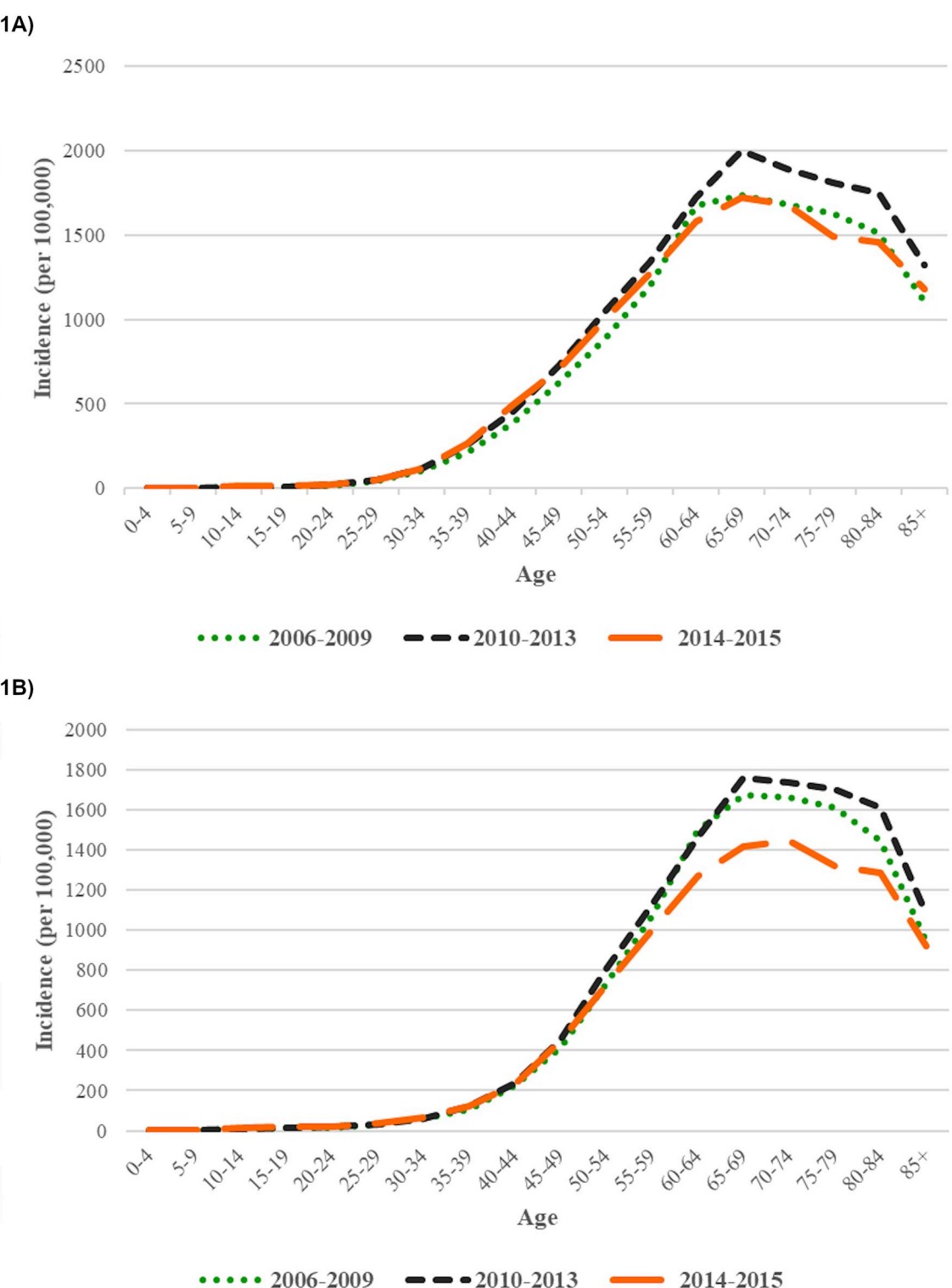

**Fig 1. Incidence of type 2 diabetes in Hong Kong Chinese men (A) and women (B) by age stratified by the following calendar periods: 2006–2009, 2010–2013, and 2014–2015.**

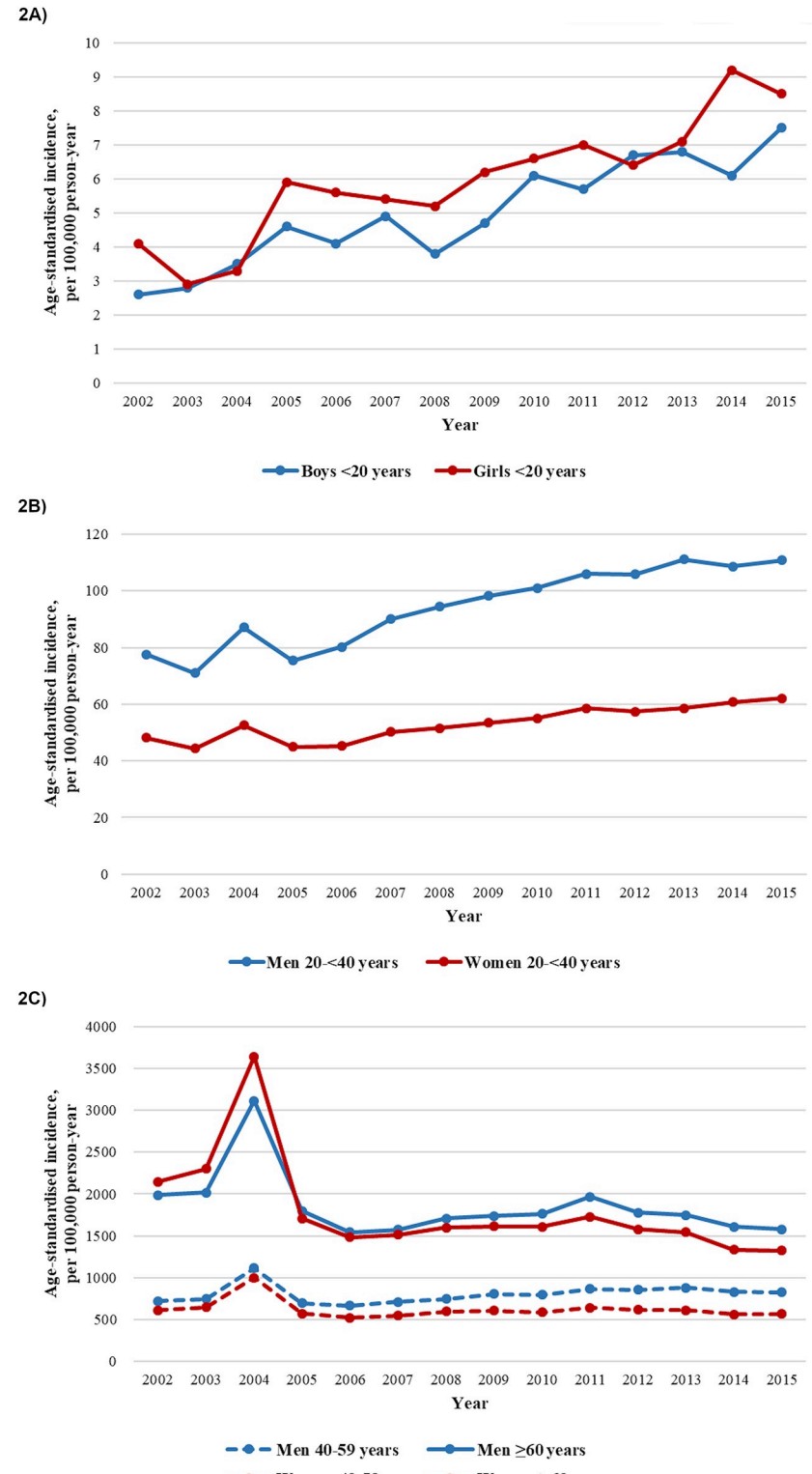

**Fig 2. Age-standardised incidence of type 2 diabetes in Hong Kong Chinese aged <20 years (A), 20 to <40 years, (B) and ≥40 years (C) between 2002 and 2015.**

2008 and 2015. In women, the incidence increased from 45.0 (95% CI 41.4–48.6) to 55.1 (95% CI 51.0–59.1) per 100,000 person-years (APC 4.6% [95% CI 2.8–6.4], $p < 0.05$) between 2005 and 2010 and from 55.1 (95% CI 51.0–59.1) to 62.1 (95% CI 57.8–66.3) per 100,000 person-years (APC 2.0% [95% CI 0.4–3.6], $p < 0.05$) between 2010 and 2015. The AAPCs were 4.2% (95% CI 3.1–5.3) and 3.3% (95% CI 2.3–4.2) in men and women, respectively, for the entire period ($p < 0.05$). The sex disparity in incidence of type 2 diabetes varied by age. For people <20 years old, the incidence was disproportionately higher in girls than in boys, with an overall rate ratio of 1.22 (95% CI 1.10–1.37, $p < 0.05$). Upon reaching young adulthood, the risk reversed, and men had higher incidence rates than women.

In people aged 40 to <60 years, we detected an initial increase in incidence followed by stabilisation in both sexes (Table 1, Fig 2C). In men, the incidence of type 2 diabetes increased from 697.0 (95% CI 681.1–712.8) to 852.7 (95% CI 836.0–869.3) per 100,000 person-years between 2005 and 2012 (APC 3.9% [95% CI 2.4–5.4], $p < 0.05$) and was constant until 2015 (APC −1.9% [95% CI −6.7 to 3.1], $p = 0.37$). In women, the incidence increased from 569.7 (95% CI 555.3–584.0) to 639.8 (95% CI 601.8–628.5) per 100,000 person-years between 2005 and 2011 (APC 2.7% [95% CI 0.1–5.3], $p < 0.05$) and had no significant change thereafter (APC −2.8% [95% CI −6.9 to 1.5], $p = 0.16$). Among people aged ≥60 years, the incidence was stable between 2005 and 2011 (APC 1.1% [95% CI −1.8 to 4.2], $p = 0.39$), followed by a decline (APC −5.8% [95% CI −10.6 to −0.8], $p < 0.05$) in women, whereas in men, the incidence of type 2 diabetes was unchanged across the observation period. Trend analyses using restricted cubic splines are presented in S3 Fig.

We determined the HbA1c levels at diagnosis of diabetes among people who were diagnosed using HbA1c criteria (S4 Fig). Between 2002 and 2011, HbA1c levels at diagnosis of diabetes decreased from 8.5% to 8.0% ($p < 0.05$ for a linear trend) in men and from 8.1% to 7.6% ($p < 0.05$ for a linear trend) in women. From 2011 to 2015, however, the HbA1c levels at diagnosis remained static in both sexes.

## Discussion

Using the territory-wide HKDSD, we conducted time trend analyses on the incidence of type 1 and type 2 diabetes in people in Hong Kong. Between 2005 and 2015, the incidence of type 2 diabetes increased in people aged <20 years and 20 to <40 years, increased and then stabilised in people aged 40 to <60 years, and was unchanged in those aged ≥60 years. Incidence of type 1 diabetes continued to climb in people aged <20 years but remained constant in other age groups. Although the incidence of type 1 diabetes was the highest in those <20 years old, adult-onset type 1 diabetes accounted for over two-thirds of newly diagnosed cases during the surveillance period. Among those <20 years old, type 2 diabetes contributed to more than half of the newly presented cases. The key strengths of this study were the use of population-level data through access to the electronic medical record system of the Hong Kong HA and its low susceptibility to selection bias. Given the scarcity of population-level data on secular changes in diabetes incidence in Asia, our results are timely in providing insights into the contemporary diabetes epidemic in this region. Importantly, this study highlights the emerging problem of young-onset diabetes and calls for effective strategies to reduce modifiable risk factors for diabetes in this group.

### Incidence of type 1 diabetes

The multinational European Diabetes (EURODIAB) register, which included 29,311 incident cases of type 1 diabetes in youth aged <15 years recorded between 1989 and 2003, revealed regional differences in the trend of diabetes incidence [8]. In areas with a high burden of type 1 diabetes, such as the Nordic countries, the United States, and Australia, the incidence

appeared to have stabilised over the past decade [9,10,11]. Conversely, the incidence of type 1 diabetes has continued to rise in places of low disease prevalence, including East Asia [12,13]. Based on a registry of 622 newly diagnosed cases of type 1 diabetes in children aged <15 years in Shanghai, Zhao and colleagues observed an increase in incidence from 1.5 per 100,000 person-years in 1997–2001 to 5.5 per 100,000 person-years in 2007–2011 [12]. From the 2012–2014 National Health Insurance Service database containing 706 physician-reported cases of type 1 diabetes in children aged <15 years in South Korea, Kim and colleagues reported an incidence of 3.2 per 100,000 person-years, which was 2.3-fold higher compared with the rate recorded in the earlier period of 1995–2000 [13]. In the present study, we also detected an increase in the incidence of type 1 diabetes in children and adolescents, and the increase occurred in both sexes. In Hong Kong, the incidence of type 1 diabetes has not been determined since the last report 2 decades ago. Based on retrospective retrieval of 255 paediatric cases of newly diagnosed diabetes between 1984 and 1996, Huen and colleagues recorded an incidence of 1.4 per 100,000 person-years for type 1 diabetes in children aged <15 years in Hong Kong, which was considerably lower than our updated estimates of 5.3–6.4 per 100,000 person-years in a comparable age group [14]. Our observations, together with others' observations showing a similar rise in incidence, remain unexplained, although environmental factors, including obesity, nutrition, climate, and infection, have been implicated in the induction and acceleration of beta cell destruction [15]. From surveys of Hong Kong school children, the prevalence of overweight or obesity rose from 11.6% in the mid-1990s to 16.7% in 2005 [16], and the prevalence was 17.6% in primary school students and 19.9% in secondary school students in the last estimation in 2018 [17]. It is noteworthy that although the incidence was the highest in the paediatric population, adults aged >20 years accounted for two-thirds of the newly diagnosed cases in our database. In contrast to the rising trend of childhood-onset type 1 diabetes, the incidence of adult-onset type 1 diabetes has not increased.

## Incidence of type 2 diabetes in youth and young adults

The growing number of young people acquiring type 2 diabetes is a major health concern that is now seen globally. In the present study, 60% of incident cases of diabetes in people aged <20 years were type 2 diabetes. The SEARCH for Diabetes in Youth registry, which systematically identified and followed youth with diabetes in the US, reported an annual increase of 4.8% in the incidence of type 2 diabetes in people aged <20 years between 2002 and 2012, from 7.0 to 9.0 per 100,000 person-years in boys and from 11.1 to 16.2 per 100,000 person-years in girls [18]. Furthermore, the increase was larger in Asians and Pacific Islanders (annual increase 16.0%) than Europeans (annual increase 3.3%). Similarly, Wu and colleagues reported an increase in incidence from 0.7 to 3.6 per 100,000 person-years using a registry of 392 newly diagnosed cases of young-onset diabetes presented between 2007 and 2013 in Zhejiang, China [19]. Among those aged <20 years in Hong Kong, the average annual increase in incidence was 4.8%–5.9%, and the last recorded crude incidence rates in 2015 were 8.3 and 9.2 per 100,000 person-years in boys and girl, respectively, which were lower than the US figures but higher than rates in China. Similarly, we detected an increase in incidence of type 2 diabetes in people aged 20 to <40 years in both sexes. Besides exposure to an increasingly obesogenic environment, intrauterine effects from gestational diabetes and/or maternal obesity and exposure to present-day endocrine disruptors may be responsible [20–22]. It is also possible that the observed incidence trend of young-onset type 2 diabetes was the result of people seeking medical attention earlier in their disease trajectory rather than an actual fall in the age of diabetes development. In this regard, people who would develop diabetes were diagnosed progressively earlier, thus removing people from the undiagnosed pool who would have otherwise presented at an older age.

## Incidence of type 2 diabetes in middle-aged to older adults

In people aged 40 to <60 years, the incidence of type 2 diabetes in both sexes initially increased and then flattened from 2011/2012 onward. In women aged ≥60 years, the incidence was stable until 2011 and declined thereafter, whereas in men aged ≥60 years, the incidence remained constant. The absence of a rise might reflect levelling off in exposure to risk factors. In Hong Kong, the prevalence of obesity has been stable in men and has declined in women since the mid-1990s [23,24], which is attributable to a series of government-led health promotion initiatives targeting healthier lifestyles, such as the EatSmart programme and mandatory nutrition labelling during this period [25]. Women are generally more receptive to health information and ready to adopt a healthy lifestyle compared with men, which might in part account for the sex difference in trends of obesity and type 2 diabetes [26]. Cigarette smoking is linked to the development of type 2 diabetes [27]. Antismoking advertising campaigns and government policy to ban smoking in many public areas have resulted in a reduction in smoking rates over the past 35 years, from 23.3% in 1982 to 10.0% in 2017 in Hong Kong [28]. Lastly, saturation effect could be an explanatory factor, wherein increased health awareness and improved screening efforts during earlier periods have captured most of the high-risk individuals, thus depleting the pool of undiagnosed type 2 diabetes over time. In support of this, we observed a significant decline in HbA1c values at diagnosis of type 2 diabetes from 2002 to 2011, possibly reflecting proactive case finding and earlier detection. Although screening programmes for diabetes have not been formally introduced in Hong Kong, governmental policies to enhance primary care, including the dissemination of a reference framework for diabetes care, could have an effect toward improving diabetes detection [29,30]. The increasing number of outpatient and inpatient attendances to public healthcare facilities over time also supports an increasing opportunity for diagnosing diabetes [31].

Limited studies indicated interethnic variation in the risks for type 2 diabetes. Alangh and colleagues reported 10-year secular trends in diabetes incidence in Ontario, Canada, and found that incidence increased moderately by 24% in the European population as compared with a 15-fold increase in the Chinese population, in which the absolute incidence rate was twice that of Europeans (19.6 versus 10.0 per 1,000 person-years) by 2005 [32]. From contemporary studies, the age- and sex-adjusted diabetes incidence rates among adults were 370 per 100,000 person-years in the United Kingdom [33], 398 per 100,000 person-years in Sweden [34], and 710 per 100,000 person-years in the US during the 2012–2014 period [35]. Over a comparable reporting period, the rates were 1,099 and 949 per 100,000 person-years in men and women in our study, 830 per 100,000 person-years in Korea [36], and 800 per 100,000 person-years in Taiwan [37]. Differences in methods used to capture incident cases and diagnostic intensities would have influenced the estimates. Allowing for these factors, available data, including those from the present study, suggest that the incidence of type 2 diabetes may be higher in East Asians than Europeans. Although Chinese people are leaner than Europeans, for the same BMI, the former have more visceral fat, greater insulin resistance, and more metabolic complications [38]. Moreover, low body weight is linked to poorer pancreatic beta cell reserve, and Chinese people are more vulnerable to external factors, such as glucolipotoxicity, that trigger progressive decline in beta cell function [39].

## Transient rise in the number of diabetes cases in 2004

A sharp peak in the incidence of type 2 diabetes was recorded in 2004, and this could be connected to the severe acute respiratory syndrome epidemic in 2003 in Hong Kong [40]. The excess of 21,000 new cases of type 2 diabetes in 2004 compared with 2003 might be due to heightened health vigilance at the aftermath of the epidemic, which prompted increased doctor visits. The modest dip in the number of new cases in the ensuing 3 years (from 2005 to 2007) compared with numbers between 2002 and 2003 could support earlier identification of cases

during the transitory period, leaving fewer cases to be detected in subsequent years. The excess might also represent an injection of prevalent cases from the private sector to the public system in response to economic adversity.

## Limitations

We acknowledge the following limitations in the study. An electronic medical record database was used to identify patients with diabetes, and we cannot exclude the possibility of incomplete capture. The data source included patients who attended only public healthcare facilities, and those receiving care in the private sector—who represent about 10% of the entire disease population—were not included. As such, the incidence rates as reported were likely underestimations of the actual figure by up to 10%. However, except for 2004, there was no evidence that the ratio of medical care in private sector versus public sector has changed over time, and therefore, this is unlikely to affect assessment of incidence trends. Two-hour plasma glucose by an oral glucose tolerance test (OGTT) was not included as one of the qualifying criteria for diabetes, which could also underestimate incidence rates, especially in the paediatric population in whom HbA1c alone has low sensitivity of diagnosing type 2 diabetes [41]. The categorisation of type 1 and type 2 diabetes was based on an algorithm rather than direct inspection of the clinical notes. Although the algorithm was developed in an independent data set in which the diagnosis of type 1 diabetes was verified by reviewing medical records, the absence of confirmatory tests such as anti-islet cell antibodies or C-peptide levels could lead to incorrect differentiation of type 1 and type 2 diabetes, affecting the accuracy of the algorithm. Misclassifying type 2 as type 1 diabetes would significantly inflate the incidence of type 1 diabetes, whereas the impact on incidence of type 2 diabetes was probably minimal. The probability of incorrect grouping would be greater in older patients in whom the algorithm performed less well, and estimates of incidence of type 1 diabetes in adults will require confirmation in other cohorts. Because of limitations in details of the extracted information, we could not discern patients with other aetiologies of diabetes, such as maturity-onset diabetes of the young and secondary diabetes. Owing to the asymptomatic nature of diabetes, diagnosis is often delayed. Despite the good quality of the surveillance database, undiagnosed cases could not be captured, leading to underreporting of case burden. The recorded incidence rates were also sensitive to secular changes in people's health-seeking behaviour, screening practice, and prescription behaviour.

## Summary

In this report on the secular trend of the incidence of diabetes in Hong Kong, we revealed that the incidence of type 1 diabetes increased in people aged <20 years and was stable in other age groups. The incidence of type 2 diabetes also increased in people aged <40 years and accounted for over half of new cases of diabetes among people aged <20 years. In people aged ≥40 years, the incidence of type 2 diabetes remained constant. These observations provide an impetus for scaling up measures to prevent development of diabetes in people at risk.

## Supporting information

**S1 Table. Number of people with incident and prevalent type 1 or type 2 diabetes in the Hong Kong Diabetes Surveillance Database, 2002–2015.**
(DOCX)

**S2 Table. Demographic and clinical characteristics of people with incident type 1 or type 2 diabetes at diagnosis in the Hong Kong Diabetes Surveillance Database, 2002–2015.**
(DOCX)

**S1 Fig. Description of the Hong Kong Diabetes Surveillance Database, 2002–2015.**
(TIF)

**S2 Fig. Number of new cases of type 2 diabetes by year in the Hong Kong Diabetes Surveillance Database, 2002–2015.**
(TIF)

**S3 Fig. Incidence trends of type 2 diabetes in men and women using restricted cubic spline, 2002–2015.** (A) Incidence trends of type 2 diabetes in boys aged <20 years using restricted cubic spline. (B) Incidence trends of type 2 diabetes in men aged 20 to <40 years using restricted cubic spline. (C) Incidence trends of type 2 diabetes in men aged 40 to <60 years using restricted cubic spline. (D) Incidence trends of type 2 diabetes in men aged ≥20 years using restricted cubic spline. (E) Incidence trends of type 2 diabetes in girls aged <20 years using restricted cubic spline. (F) Incidence trends of type 2 diabetes in women aged 20 to <40 years using restricted cubic spline. (G) Incidence trends of type 2 diabetes in women aged 40 to <60 years using restricted cubic spline. (H) Incidence trends of type 2 diabetes in women aged ≥20 years using restricted cubic spline.
(TIF)

**S4 Fig. Mean HbA1c levels at diagnosis in men and women with incident type 2 diabetes between 2002 and 2015.** HbA1c, glycated haemoglobin.
(TIF)

**S1 STROBE Checklist. STROBE, Strengthening the Reporting of Observational Studies in Epidemiology.**
(DOC)

## Acknowledgments

We acknowledge the Hong Kong Hospital Authority for providing the data for research.

## Author Contributions

**Conceptualization:** Andrea O. Y. Luk, Calvin Ke, Eric S. H. Lau, Hongjiang Wu, William Goggins, Ronald C. W. Ma, Elaine Chow, Alice P. S. Kong, Wing-Yee So, Juliana C. N. Chan.

**Data curation:** Andrea O. Y. Luk, Calvin Ke.

**Formal analysis:** Calvin Ke, Eric S. H. Lau, Hongjiang Wu.

**Methodology:** Andrea O. Y. Luk, Calvin Ke, Eric S. H. Lau.

**Writing – original draft:** Andrea O. Y. Luk.

**Writing – review & editing:** Andrea O. Y. Luk, Calvin Ke, Eric S. H. Lau, Hongjiang Wu, William Goggins, Ronald C. W. Ma, Elaine Chow, Alice P. S. Kong, Wing-Yee So, Juliana C. N. Chan.

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
