## [Decision Letter · Decision Letter 0]

7 Sep 2019

Dear Dr. Luk,

Thank you very much for submitting your manuscript "Trends on incidence of type 1 and type 2 diabetes in Hong Kong, China: Analysis of the Hong Kong Diabetes Surveillance Database" (PMEDICINE-D-19-01907) for consideration at PLOS Medicine. 

[LINK]

In light of these reviews, I am afraid that we will not be able to accept the manuscript for publication in the journal in its current form, but we would like to consider a revised version that addresses the reviewers' and editors' comments. Obviously we cannot make any decision about publication until we have seen the revised manuscript and your response, and we plan to seek re-review by one or more of the reviewers. 

We expect to receive your revised manuscript by Sep 23 2019 11:59PM. Please email us (plosmedicine@plos.org) if you have any questions or concerns.

We look forward to receiving your revised manuscript. 

Sincerely,

Clare Stone, PhD, Acting Editor-in-Chief

for 

Adya Misra

Senior Editor

PLOS Medicine

plosmedicine.org

We note there are some substantial issues to be resolved as highlighted by the referees. Please also address the following points:

Title – I wonder if a clearer study design could be used for the 2nd part of the title?

Abstract – there should be 3 sections: Background, Methods and Findings, Conclusions- please recast; Please quantify any results related to data with 95%Cis and p values; please provide a sentence on the limitations of the study as the final sentence of the ‘Methods and Findings section; please do add some summary level demograhic information; 

Data – you say that no access is allowed – not even for users who meet requirements? 

Please use square brackets for refs in the main text. 

Line 113 – type 1 or 2? 

Line 167 – please define diabetes codes

Line 177- define how diabetes is defined – what measure?

Line 199- I am confused by this sentence. Please clarify ‘Of 778,051 people captured in the HKDSD, we excluded 33,916 people with diabetes codes but not fulfilled other criteria of diabetes.’

Throughout you tend to fall back on describing simply diabetes, as this paper is about type 1 and 2, please be careful to always state which(or if both)

The word youth seems a little odd, please instead say people under 20 years of age

Figure 1 – can I suggest you use different colours instead of 3 variations of blue

Please use the "Vancouver" style for reference formatting, and see our website for other reference guidelines https://journals.plos.org/plosmedicine/s/submission-guidelines#loc-references. For example 6 names and then et al.

Mian text – is it possible to provide some information perhaps in table for – adding to one of the current tables – with more participant characteristics, eg BMI, married, smoking etc…

Please provide a STROBE reporting checklist 

Do you have a prespecified analysis plan, if so please provide as a Supp file and a call out in the methods to it. If no plan exists, please state when the analyses were planned in relation to data analysis. 

Comments from the reviewers:

Reviewer #1: I confine my remarks to statistical aspects of this paper. Unfortunately, I have some major issues that need to be resolved before I can recommend publication.

First, joinpoint analysis needs to be described and its use needs to be justified. It's a fairly obscure method and I had trouble finding literature describing it, but it appears to be a restricted linear spline method. However, while I do like the use of spline models, there are reasons to prefer restricted cubic splines. And spline methods are a) Well known and b) Implemented in popular software including R and SAS, rather than relying on specialized software.

Second, the results of the regression analysis don't seem to be included. Are the figures the results of the analysis? Are there other results? What about parameter estimates?

Third, categorizing age is almost surely a mistake. Use age as "number of years".

More specific comments:

Line 169 - 173 Something very strange is happening here. Perfect results for under 20 and then really bad results for over 20 make me suspicious that something is going wrong somewhere or, perhaps there are very few people under 20 with type I diabetes, which could lead to overfitting. The authors note that 103 people in HKDR had type I and it seems like about 1/3 were under 20 (although this doesn't seem to be explicitly stated). 

Line 210 and other places. The authors state that many trends were linear, apparently from testing a model that had year as a linear effect on rate. But the fact that a linear trend is statistically significant does not mean the trend was linear, it just just means a straight line is a better fit than a null model. The tables and the graphs show that the trends were not linear at all - they went up and down in a complex pattern. 

Table 1 may not be needed. It seems to duplicate the material from the graphs, which are easier to interpret. If Table 1 is kept, then the AAPC and the p value should be removed - they are misleading. 

The age categories for the figures don't match the age caegories for the table - it's unclear why not, or what was actually included in the model - age as 0-20, 20-40 etc or age as 0-4, 5-9 etc or age as integer or what.

Peter Flom

Reviewer #2: This is a well written and clear paper that is of importance understanding diabetes epidemiology in this region of the world.

I have a number of recommendations to improve the manuscript.

1. Study population line 129. Please comment in limitations section further about diagnosing diabetes by HbA1c alone in chidlren. Please see Nowicka P, Santoro N, Liu H et al (2011) Utility of hemoglobin A(1c) for diagnosing prediabetes and diabetes in obese children and adolescents. Diabetes Care 34: 1306-11

2. Study population line 135. Please comment about why 1 day was chosen, is this sufficient time for a diagnosis of diabetes to be made e.g. revised diagnosis and subsequently drugs quickly stopped?

3. Study population line 159. Please comment specifically in the limitations section about the lack of islet cell antibodies for diagnosising T1DM in children. This is often crucial to distinguish T1 from T2 in children especially in absence of c peptide/insulin and in a population of T1 who may be increasingly overweight. Please see diagnosistic criteria used for T2DM in this recent study Candler TP, Mahmoud O, Lynn RM et al (2018a) Continuing rise of type 2 diabetes incidence in children and young people in the UK. Diabet Med 35: 737-44.

4. Figure 2 needs revising to better display the rising incidence in <20 year olds. In the current graph, the scale for this age group is not useful to see any change in trend. Suggest split figures for >20 and <20 to better show change incidence. This is a major finding in your study but is not displayed clearly in the results.

5. Line 285. Please expand further in the discussion as the cause of a reduction in incidence in women >60, can the authors speculate or suggest from other studies a cause for this?

6. Line 293. Please comment further regarding lower HbA1c at diagnosis across the time period 2002-2011 - is this statistically significant i.e. can you provide a p value/statistical test? Was there a specific public health campaign across this period suggesting people present earlier or a screening programme introduced? Please speculate as the the cause of this further in the discussion.

Reviewer #3: The author used national wide data to estimate trends of incidence of diabetes types. They could also look into prevalence of diabetes. 

The authors presented data in adults and younger groups, it would be good to separately present data in adults and young adults and children

What are key drivers for such trends? The authors might have access to clinical information (e.g. BMI) and other sociodemographic data to investigate these?

Also it would be useful to provide diabetes incidence projection and possibly to calculate the burden attributable to diabetes (DALY) in this country.

[LINK]

---

## [Decision Letter · Decision Letter 1]

18 Dec 2019

Dear Dr. Luk,

Thank you very much for re-submitting your manuscript "Secular trends on incidence of type 1 and type 2 diabetes in Hong Kong" (PMEDICINE-D-19-01907R1) for review by PLOS Medicine.

I have discussed the paper with my colleagues and the academic editor and it was also seen again by xxx reviewers. I am pleased to say that provided the remaining editorial and production issues are dealt with we are planning to accept the paper for publication in the journal.

[LINK]

We look forward to receiving the revised manuscript by Dec 23 2019 11:59PM. 

Sincerely,

Clare Stone, PhD

Managing Editor 

PLOS Medicine

plosmedicine.org

Requests from Editors:

Title- in order to adhere to PLOS Medicine style guide, your title must contain a study descriptor. Please revise to “Secular trends on incidence of type 1 and type 2 diabetes in Hong Kong: a retrospective cohort study”

Data availability- The Data Availability Statement (DAS) requires revision. For each data source used in your study: a) If the data are freely or publicly available, note this and state the location of the data: within the paper, in Supporting Information files, or in a public repository (include the DOI or accession number). b) If the data are owned by a third party but freely available upon request, please note this and state the owner of the data set and contact information for data requests (web or email address). Note that a study author cannot be the contact person for the data. c) If the data are not freely available, please describe briefly the ethical, legal, or contractual restriction that prevents you from sharing it. Please also include an appropriate contact (web or email address) for inquiries (again, this cannot be a study author).

Abstract- 95% CI is okay, we do not need the words “confidence interval” as this is a standard term 

Abstract line 81 should be “subject” not “subjected”

Abstract conclusions- both sentences are very similar and there is no indication what is meant by youth, young adults or older adults. Please specify age groups and could it be rephrased to “There was an increase in incidence of both Type 1 and Type 2 diabetes between the age of _ and _ but remained stable in the age group __ ”. 

Author summary please reword “over half is coming from Asia” to “over half of the diabetes population come from Asian countries” or similar 

Author summary- please remove “ calls for preventative actions …” as your study determined the trends and the significance of your findings needs a better explanation

Introduction- I don’t think we have a Type 2 diabetes pandemic. I appreciate the rates are increasing and there are public health concerns but please tone down.

Introduction- please change “over-weight” to “overweight”

Methods- please mention earlier in the methods section whether your study had a prospective analysis plan. If not, please provide reasons

Page 24 Line 414- please remove the word “destined” and consider more appropriate language for a scientific article 

Page 24 please provide a reference to support “women are generally more receptive…”

Please ensure you use consistent terminology for ethnicity throughout the text- such as “east Asian” or “Chinese” as they are both used interchangeably. 

Page 27, please consider revising the limitation section to remove the words “firstly” “fifthly” etc as these are not appropriate for a research article 

Throughout the text, when you say older or younger adults please add the age group referred to for additional clarity 

Line 161 – does this need updating? 250.xx

Line 429 “marked reduction” – please be more specific and give a %decline

The STROBE checklist cannot contain page or line numbers as these change during the publication process. Please reference the appropriate paragraphs/sections 

Comments from Reviewers:

Reviewer #1: The authors have addressed my concerns and I now recommend publication.

Peter Flom

[LINK]

---

## [Editor Report · Decision Letter 2]

22 Jan 2020

Dear Dr Luk, 

On behalf of my colleagues and the academic editor, Dr. Sanjay Basu, I am delighted to inform you that your manuscript entitled "Secular trends in incidence of type 1 and type 2 diabetes in Hong Kong: a retrospective cohort study" (PMEDICINE-D-19-01907R2) has been accepted for publication in PLOS Medicine. 

PRODUCTION PROCESS

PRESS

PROFILE INFORMATION

Thank you again for submitting the manuscript to PLOS Medicine. We look forward to publishing it. 

Best wishes, 

Adya Misra, PhD

Senior Editor

PLOS Medicine

plosmedicine.org